# Efficacy of Lenvatinib Combined with Transcatheter Intra-Arterial Therapies for Patients with Advanced-Stage of Hepatocellular Carcinoma: A Propensity Score Matching

**DOI:** 10.3390/ijms241813715

**Published:** 2023-09-05

**Authors:** Shigeo Shimose, Hideki Iwamoto, Takashi Niizeki, Masatoshi Tanaka, Tomotake Shirono, Etsuko Moriyama, Yu Noda, Masahito Nakano, Hideya Suga, Ryoko Kuromatsu, Takuji Torimura, Hironori Koga, Takumi Kawaguchi

**Affiliations:** 1Division of Gastroenterology, Department of Medicine, Kurume University School of Medicine, Kurume 830-0011, Japan; iwamoto_hideki@med.kurume-u.ac.jp (H.I.); niizeki_takashi@kurume-u.ac.jp (T.N.); shirono_tomotake@med.kurume-u.ac.jp (T.S.); moriyama_etsuko@med.kurume-u.ac.jp (E.M.); noda_yuu@med.kurume-u.ac.jp (Y.N.); nakano_masahito@kurume-u.ac.jp (M.N.); ryoko@med.kurme-u.ac.jp (R.K.); hirokoga@med.kurume-u.ac.jp (H.K.); takumi@med.kurume-u.ac.jp (T.K.); 2Iwamoto Internal Medical Clinic, Kitakyusyu 802-0832, Japan; 3Clinical Research Center, Yokokura Hospital, Miyama 839-0295, Japan; mazzo6528@me.com; 4Department of Gastroenterology and Hepatology, Yanagawa Hospital, Yanagawa 832-0077, Japan; suga516@med.kurume-u.ac.jp; 5Department of Gastroenterology and Hepatology, Omuta City Hospital, Omuta 836-8567, Japan; tori@med.kurume-u.ac.jp

**Keywords:** lenvatinib, TACE, HAIC, advance-stage

## Abstract

This study aimed to evaluate the effect of lenvatinib (LEN) combined with transcatheter intra-arterial therapy (TIT) for advanced-stage hepatocellular carcinoma (HCC) after propensity score matching (PSM). This retrospective study enrolled 115 patients with advanced-stage HCC who received LEN treatment. The patients were categorized into the LEN combined with TIT group (*n* = 30) or the LEN monotherapy group (*n* = 85). After PSM, 38 patients (LEN + TIT group, *n* = 19; LEN monotherapy group, *n* = 19) were analyzed. The median overall survival (OS) in the LEN + TIT group was significantly higher than that in the LEN monotherapy group (median survival time (MST): 28.1 months vs. 11.6 months, *p* = 0.014). The OS in the LEN combined with transcatheter arterial chemoembolization and LEN combined with hepatic arterial infusion chemotherapy groups was significantly higher than that in the LEN monotherapy group (MST 20.0 vs. 11.6 months, 30.2 vs. 11.6 months, *p* = 0.048, and *p* = 0.029, respectively). Independent factors associated with OS were alpha-fetoprotein and LEN combined with TIT. The indications for LEN combined with TIT were age <75 years and modified albumin bilirubin (m-ALBI) grade 1. We concluded that LEN combined with TIT may improve prognosis compared with LEN monotherapy in patients with advanced-stage HCC.

## 1. Introduction

Hepatocellular carcinoma (HCC) is a leading cause of cancer-related deaths, and its incidence is increasing worldwide, including in Asia [1]. Currently, the Barcelona Clinic Liver Cancer (BCLC) classification is the most widely used for predicting prognosis and determining treatment modalities for patients with HCC [2]. Although early-stage HCC can often be effectively treated through liver resection or radiofrequency ablation when detected early [3,4], patients are frequently diagnosed at an intermediate or advanced stage. As a result, several patients with unresectable HCC (u-HCC) undergo systemic therapies [1]. However, the prognosis for these patients remains unsatisfactory. Based on the findings of the REFLECT trial, lenvatinib (LEN) is a molecular target agent (MTA) approved for first-line treatment [5] and widely utilized in the management of patients with u-HCC. LEN specifically targets vascular endothelial growth factor receptors (VEGF) 1–3 and fibroblast growth factor receptors (FGFR) 1–4, offering the potential for a higher response rate compared to other MTAs [6]. However, as the therapeutic effect of LEN monotherapy in patients with u-HCC is limited, there is growing interest in exploring the strategy of combining LEN with other therapies [7,8]. We previously reported that the combination of LEN with transcatheter intra-arterial therapies (TIT), such as transcatheter arterial chemoembolization (TACE) and hepatic arterial infusion chemotherapy (HAIC), improved the prognosis of patients with intermediate-stage HCC [9]. Moreover, several observational studies have reported that sequential therapy with LEN and TACE provides a survival benefit following the progression of u-HCC [10,11]. Fang et al. reported improved survival outcomes in patients with HCC treated with LEN combined with HAIC [12]. While the efficacy of LEN combined with TIT, targeting intrahepatic tumors, has been demonstrated in intermediate-stage HCC, a recent clinical trial has shown its effectiveness even in advanced-stage HCC [13]. The LAUNCH trial revealed improved clinical outcomes in patients with advanced-stage HCC treated with LEN combined with TACE [13]. Currently, the progression of intrahepatic lesions is the leading cause of death in most patients with advanced-stage HCC [14]. Therefore, controlling the status of intrahepatic tumors is considered an important predictor of survival, even in patients with advanced-stage HCC [15,16,17]. However, the beneficial effects of LEN combined with TIT on survival in patients with advanced-stage HCC in real-world clinical practice remain unclear. Therefore, this study aimed to evaluate the efficacy of LEN combined with TIT in patients with advanced-stage HCC using propensity score matching analysis (PSM).

## 2. Results

### 2.1. Characteristics of Patients

The characteristics of the 115 patients with advanced-stage HCC are summarized in Table 1. The median age was 71 (38–89). Among the patients, 36.5% (42/115) had non-hepatitis B or C viruses as the etiology of liver diseases. Modified ALBI (m-ALBI) grade 1 was observed in 47 patients (40.8%), m-ALBI grade 2a in 34 patients (29.6%), and m-ALBI grade 2b in 34 patients (29.6%). MVI was present in 28 patients (24.3%) and extrahepatic spread (EHS) in 88 patients (76.5%). LEN combined with TIT was administered to 26.1% (30/115) of the patients, with 12 patients receiving TACE and 12 patients receiving HAIC. Regarding treatment lines, 77 patients (66.9%) received first-line treatment, 27 patients (23.5%) received second-line treatment, and 11 patients (9.6%) received third-line treatment. While age, ALBI score, m-ALBI grade, and macrovascular invasion showed significant differences between the LEN combined with TIT and LEN monotherapy groups, no other significant differences were observed.

### 2.2. Therapeutic Outcomes of LEN according to mRECIST

The objective response rate (ORR) and disease control rate (DCR) according to mRECIST, the ORR and DCR were 26.1% (30/115) and 78.7% (79/115), respectively (Appendix A, Table A1).

### 2.3. Overall Survival with LEN for the Advanced Stage before PSM

The median survival time (MST) was 12.8 months (Appendix A, Figure A1).

### 2.4. Comparison of OS between LEN Combined with TIT and LEN Monotherapy before PSM

The OS in the LEN combined with the TIT group was significantly higher than that in the LEN monotherapy group (MST 28.1 months vs. 11.4 months, *p* < 0.001) (Figure 1A). Although there was no difference in OS between the LEN + TACE and LEN + HAIC groups (MST 23.1 vs. 36.0 months, *p* = 0.578), the OS in the LEN + TACE and LEN + HAIC groups was significantly higher than that in the LEN monotherapy group (*p* = 0.031 and *p* = 0.001, respectively) (Figure 1B).

### 2.5. Patient Characteristics after PSM

To minimize the influence of confounding factors, PSM was conducted considering the following variables: age, sex, etiology, ALBI score, tumor size, tumor number, EHS, MVI, and AFP. After matching, there were no significant differences in age, ALBI score, or MVI between the LEN combined with TIT and LEN monotherapy groups (Table 2).

### 2.6. Therapeutic Outcomes of LEN according to RECIST after PSM

The therapeutic responses of the two groups are shown in Table 3. There were significant differences in ORR and DCR between the two groups (47.3% vs. 15.7%, *p* = 0.032; 94.7% vs. 63.1%, *p* = 0.017, respectively).

### 2.7. Comparison of OS between LEN Combined with TIT and LEN Monotherapy after PSM

The OS in the LEN combined with TIT group was significantly higher than that in the LEN monotherapy group (MST 23.1 months vs. 11.6 months, *p* = 0.014) (Figure 2A). Although there was no difference in OS between the LEN + TACE and LEN + HAIC groups (MST 20.0 vs. 30.2 months, *p* = 0.705), the OS in both the LEN + TACE and LEN + HAIC groups was significantly higher than that in the LEN monotherapy group (*p* = 0.048 and *p* = 0.029, respectively) (Figure 2B).

### 2.8. Variables Associated with Overall Survival after PSM

In the univariate analysis, tumor size, LEN combined with TIT, and AFP levels were included as variables. The multivariate analysis identified low AFP levels (<200 ng/mL) and LEN combined with TIT as independent predictors of better survival (Table 4).

### 2.9. Swimmer Plot Analysis in Patients Treated with LEN Combined with TIT

The swimmer plot for patients treated with a combination of LEN and TIT after PSM is shown in Figure 3. Nineteen patients were treated with LEN combined with TIT, and seventeen patients discontinued treatment based on the cutoff data. The median LEN treatment duration was 10.8 months (Figure 3).

### 2.10. Treatment Duration of LEN with or without TIT after PSM

The median treatment duration of LEN in the LEN combined with TIT group was significantly longer than that in the LEN monotherapy group (treatment duration 14.0 months vs. 4.4 months, *p* < 0.001) (Appendix A, Figure A2).

### 2.11. Additional Treatments after the Discontinuation of LEN after PSM

Until the time of study cessation, there were no significant differences in additional treatment rates between the LEN combined with TIT and LEN monotherapy groups (64.7% vs. 73.6%, *p* = 0.559) (Appendix A, Table A2).

### 2.12. Decision-Tree Analysis for LEN Combined with TIT

In this study, the rate of LEN combined with TIT in all subjects was 26.1% at the time of study cessation. A decision-tree analysis was performed to determine the profiles associated with LEN combined with TIT. Age was identified as the first split variable for the rate of LEN combined with TIT. Among patients aged ≥75, the rate of LEN combined with TIT was only 0.3%, while among patients aged <75, the rate of LEN combined with TIT was 37.5%. Within the group of patients aged <75 years, the m-ALBI grade was selected as the second split, and the rate of LEN combined with TIT was 50% among patients with m-ALBI grade 1 (Figure 4).

### 2.13. Comparison of AEs between the Two Groups (Grade ≥ 3)

Severe AEs, defined as ≥3 as determined by the attending physician, are shown in Appendix A, Table A3. The most frequent AE in the LEN combined with TIT group was a hand-foot-skin reaction, observed in 10.0% (3/30) of patients. In contrast, hypertension occurred in 10.5% (9/85) of patients in the LEN monotherapy group. However, no significant differences were observed between the two groups (Appendix A, Table A3).

## 3. Discussion

In this study, we have shown that AFP levels and the combination of LEN with TIT were significant predictors of OS in patients with advanced-stage HCC after PSM. Our findings suggest that patients aged <75 years and with m-ALBI grade 1 should be considered for aggressive treatment with LEN combined with TIT.

The present study revealed that the ORR and DCR were 26.1% and 78.7% according to mRECIST, and 12.2% and 78.7% according to RECIST, respectively, in patients with advanced-stage HCC treated with LEN. These therapeutic effects of LEN are consistent with those reported in a Phase III randomized clinical trial [13], indicating the standard efficacy of LEN in the enrolled subjects of our study.

In this study, we have demonstrated that low AFP levels and the combination of LEN with TIT were independent prognostic factors in patients with advanced-stage HCC after PSM. Alpha-fetoprotein (AFP) is a commonly used serum biomarker for assessing HCC management, tumor progression, and treatment response [18,19,20]. Previous studies have shown that elevated AFP values are associated with higher tumor burden, invasiveness, HCC recurrence, and poor prognosis across all stages of HCC [2,21].

Recent studies have also highlighted the potential benefits of combining TIT and LEN in improving clinical outcomes for patients with u-HCC [7,11,22]. Although anti-angiogenic drugs, including LEN, are generally effective for proliferating cancer cells depending on the blood supply, there are many reports regarding the induction of acquired resistance and various genetic changes in cancer cells with LEN treatment [23,24]. In particular, cancer stem cells can survive an ischemic change due to anti-angiogenic drugs because they are independent of the blood supply [24]. Additionally, cancer stem cells are capable of self-renewal and multipotent differentiation. Recently, cancer stem cells have been shown to contribute to vasculogenic mimicry, which is a newly defined pattern of tumor blood supply [25]. Vasculogenic mimicry provides a special passage without endothelial cells and is conspicuously different from VEGF-inducing angiogenesis and vasculogenesis. Guo et al. also demonstrated that YRDC, which was capable of binding transfer RNA (tRNA), was related to acquired resistance to LEN through regulation of the protein translation of KRAS through the modification of tRNA [26]. We also previously reported the acquired resistance mechanism of LEN due to the insulin-like growth factor-binding protein-1 (IGFBP-1) protein [27]. IGFBP1 is promoted by LEN-induced ischemia, and it could induce neo-angiogenesis in a VEGF-independent manner. On the other hand, it is also well known that LEN improves drug delivery of anticancer drugs within the tumor by improving the intratumoral microenvironment of HCC tumors by normalizing blood vessels and reducing interstitial pressure [28,29]. Therefore, TIT with improved drug delivery of anticancer drugs by LEN may directly and more effectively treat remnant LEN-resistant cancer cells. However, TIT increases tumor hypoxia, thereby leading to the upregulation of hypoxia-inducible factor-1 (HIF-1). Elevated HIF-1 upregulates the expression of VEGF and PDGF, promoting tumor revascularization and progression [30,31]. To solve these problems, re-administration of LEN after TIT prevents recurrence through the control of the residual tumor and improvement of the hypoxic condition, resulting in the inhibition of releasing hypoxia-inducible cytokines. Thus, the synergistic effect of LEN and TIT may have improved OS compared to LEN alone.

Notably, in the LEN combined with HAIC therapy group, the MST was 30.2 months, despite 75.0% of the patients having MVI. MVI is a critical condition that directly impacts liver function and is associated with a poor prognosis and shorter survival time [32,33]. Previous studies have shown promise for using HAIC for patients with MVI [34,35]. Moreover, a meta-analysis has indicated that combining molecular-targeted agents with HAIC can improve OS compared to using molecular-targeted agents alone in patients with advanced-stage HCC [36,37]. This suggests that the addition of LEN to HAIC may be a potential therapeutic strategy for patients with MVI. However, assessing the effectiveness of LEN combined with HAIC remains challenging. Additional large-scale randomized controlled trials are required to further investigate these clinical benefits.

Our findings indicate that the administration of LEN combined with TIT in patients with advanced-stage HCC is associated with age <75 years and m-ALBI grade 1, according to the decision tree analysis. Previously, we reported that younger patients with intermediate-stage HCC could undergo LEN combined with TIT [9]. It is known that older age is associated with increased vulnerability to disease and treatment [37], which suggests that older patients may experience a greater physical burden when undergoing TIT immediately after LEN treatment. Moreover, several studies have shown that preserved liver function is associated with a longer treatment duration of systemic therapy [38,39] and that treatment duration is correlated with OS [38]. In this study, the treatment duration in the LEN combined with the TIT group was significantly higher than that in the LEN monotherapy group (median treatment duration: 14.0 months vs. 4.4 months, *p* = 0.001, respectively) (Figure A2). The shorter treatment duration in the LEN monotherapy group may be attributed to the fact that patients with advanced-stage HCC often have compromised hepatic function. Therefore, younger age and preserved hepatic function may be important factors to consider when determining whether to administer LEN combined with TIT for advanced-stage HCC.

This study has some limitations. First, it was a retrospective study. Second, there was selection bias in the classification of the LEN combined with TIT and LEN monotherapy groups. Third, the decision to add on-demand TIT was made by the attending physicians, introducing potential bias. Fourth, we could not provide enough experimental evidence. Thus, a randomized, controlled, prospective validation study is required to determine the efficacy of LEN combined with TACE and HAIC in a larger number of patients with advanced-stage HCC.

## 4. Materials and Methods

### 4.1. Study Design and Patients

This multicenter retrospective cohort included 308 patients with u-HCC who received LEN treatment between 24 March 2018 and 28 February 2023, at five Japanese institutions. The cutoff date for this analysis was 31 May 2023. The study employed the following eligibility criteria: (i) age > 18 years, (ii) Eastern Cooperative Oncology Group performance status (PS) < 2, (iii) Child–Pugh class A, and (iv) complete availability of clinical data and follow-up data until the end of the study period (31 May 2023) or death. Exclusion criteria were as follows: (i) Child–Pugh class B or C; (ii) BCLC stage 0, A, or B; and (iii) a history of allergy to iodine contrast media. A total of 193 patients were excluded, leaving 115 patients who were classified into two groups: the LEN combined with TIT group (*n* = 30) or the LEN monotherapy group (*n* = 85) (Figure 1). After PSM, the efficacy analysis included 19 patients in the LEN combined with TIT group and 19 patients in the LEN monotherapy-treated group (Appendix A, Figure A3). The study was conducted in accordance with the Declaration of Helsinki and approved by the Ethics Committee of Kurume University School of Medicine (approval code: 18146). Informed consent from patients was obtained using an opt-out approach.

### 4.2. Lenvatinib Treatment Protocol

LEN (Eisai Co. Ltd., Tokyo, Japan) was administered orally to patients with u-HCC. The recommended oral dosage of LEN was 12 mg/day for patients with a body weight of ≥60 kg or 8 mg/day for patients with a body weight of <60 kg, as per the manufacturers’ instructions. However, a modified treatment protocol for LEN was employed, utilizing a 5 days on/2 days off administration schedule (i.e., the weekends-off protocol) [40], and the dose was adjusted based on the patient’s condition as assessed by the physicians. In cases where patients experienced grade ≥3 severe adverse events (AEs), the LEN dose was reduced or the treatment was temporarily halted.

### 4.3. Evaluation of Therapeutic Response and Safety

Therapeutic response was evaluated by performing dynamic computed tomography or magnetic resonance imaging every 4–6 weeks after the start of treatment, according to the modified response evaluation criteria in solid tumors (mRECIST) [41]. The Common Terminology Criteria for Adverse Events version 5.0 was adopted to assess AEs [42].

### 4.4. LEN Combined with Transcatheter Intraarterial Therapies

In cases in which a therapeutic response to LEN treatment was observed, the treatment was continued. However, for patients who experienced disease progression following LEN treatment, especially when intrahepatic lesions were negatively correlated with patient survival [14,43], we recommend the administration of LEN combined with TIT. LEN was discontinued 2 days prior to the administration of TIT. Within 2 weeks after TIT, LEN was resumed at the same dose or half the dose, depending on the patient’s condition. In terms of TIT selection, HAIC was administered to patients who were deemed unsuitable for TACE, including those with multinodular or invasive growth types, or macrovascular invasion (MVI) [44].

### 4.5. TACE

TACE was conducted using the following procedure: angiography was performed through the celiac and common hepatic arteries using a 3 or 4 Fr catheter, and digital subtraction angiography was carried out employing a nonionic iodine contrast agent. After assessing the tumor-contained segment using imaging techniques such as cone-beam computed tomography, a 1.7 or 1.9 Fr microcatheter (Piolax Inc., Kanagawa, Japan) was inserted towards the tumor-feeding artery. TACE was performed using 20–50 mg of epirubicin (Sawai Pharmaceutical Co., Ltd., Osaka, Japan) or cisplatin (Nippon Kayaku Co., Ltd., Tokyo, Japan) with lipiodol (Guerbet Co., Ltd., Tokyo, Japan), based on the size and number of tumors, in addition to absorbable gelatin sponge particles (Nippon Kayaku Co., Ltd.) [45]. Subsequent TACE procedures were repeated, as determined by the investigators.

### 4.6. HAIC

To implant the HAIC catheter, a 5-Fr-W-spiral catheter (Piolax) was inserted through the right femoral artery. The catheter was positioned with its distal end extending into the hepatic or gastroduodenal artery, and a subcutaneous port (Sofa Port, Nipro Pharma Corporation, Osaka, Japan) was implanted in the front femoral region [46]. For the HAIC regimen, a cisplatin-lipiodol suspension was prepared by suspending 50 mg of fine-powdered cisplatin in 5–10 mL of lipiodol, with the amount determined based on the tumor volume. On day 1, the cisplatin–lipiodol suspension was injected through the implanted catheter under angiography, followed by a 250 mg dose of 5-FU. Subsequently, a continuous infusion of 1250 mg 5-FU was continuously administered for 5 days using an infusion balloon pump (SUREFUSER PUMP; Nipro Pharma Corporation, Osaka, Japan).

### 4.7. The Decision-Tree Algorithm

A decision-tree algorithm was constructed to elucidate the relationship between the profiles and the administration of LEN combined with TIT, following the guidelines provided in the R software package (R-4.3.1), as previously described [47]. Data mining methods were utilized to identify the profiles.

### 4.8. Propensity Score Matching

PSM is a method that addresses the variability in covariate distributions among individuals assigned to specific interventions. It is generated by considering potential covariates that may influence group allocation [48]. In this study, propensity scores were estimated for all patients using a logistic regression model, taking into account the following baseline characteristics: age, sex, etiology, albumin-bilirubin (ALBI) score, tumor size, tumor number, extrahepatic spread, macrovascular invasion, and alpha-fetoprotein (AFP). To create comparable groups, a one-to-one nearest-neighbor matching algorithm with an optimal caliper of 0.2 was applied without replacement, resulting in 19 pairs of patients. Given the potential bias introduced by population size, the propensity score matching results were also presented as effect sizes: |value| < 0.2 indicated a negligible difference, |value| < 0.5 indicated a small difference, |value| < 0.8 indicated a moderate difference, and any other value indicated a large difference. The c-statistics were 0.84 (Appendix A, Figure A4).

### 4.9. Statistical Analysis

All data were presented as numbers or medians with ranges. Statistical analyses were carried out using JMP software (JMP Pro version 15, SAS Institute Inc., Cary, NC, USA). Statistical significance was defined as *p* < 0.05. Overall survival (OS) was calculated using the Kaplan–Meier method and assessed using the log-rank test or the Bonferroni method. Between-group comparisons were conducted using the Mann–Whitney U test and nonparametric analysis of variance. If the one-way analysis of variance yielded a significant result, differences between individual groups were examined using Fisher’s least significant difference test. The treatment duration of LEN and TIT, as evaluated by the investigators, was depicted using a Swimmer plot, as previously described [9]. Univariate and multivariate analyses were performed using the Cox proportional hazards model to identify risk factors associated with OS.

## 5. Conclusions

In conclusion, we have demonstrated that LEN combined with TIT improves the prognosis compared to LEN monotherapy in patients with advanced-stage HCC after PSM. Moreover, our findings suggest that patients below 75 years of age and those with m-ALBI grade 1 may be suitable candidates for intervention with LEN combined with TIT.

## Figures and Tables

**Figure 1 ijms-24-13715-f001:**
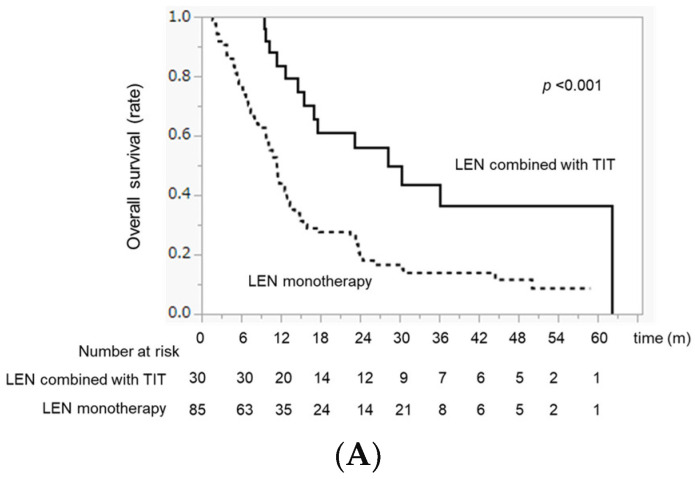
OS differences between LEN combined with the TIT group and LEN. Monotherapy group before PSM. (**A**) Median OS in the LEN combined with TIT and LEN monotherapy groups. (**B**) Median OS in the LEN combined with TACE, LEN combined with HAIC, and LEN monotherapy groups, respectively.

**Figure 2 ijms-24-13715-f002:**
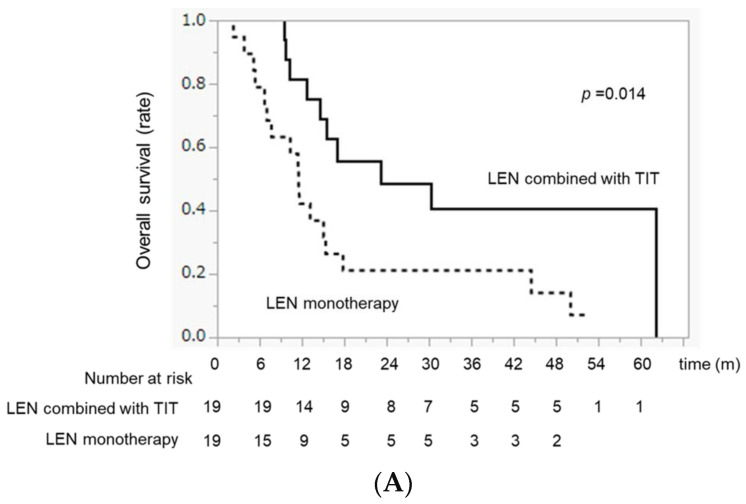
OS differences between the LEN combined with TIT group and the LEN monotherapy. group after PSM. (**A**) Median OS in the LEN combined with TIT and LEN monotherapy groups. (**B**) Median OS in the LEN combined with TACE, LEN combined with HAIC, and LEN monotherapy groups, respectively.

**Figure 3 ijms-24-13715-f003:**
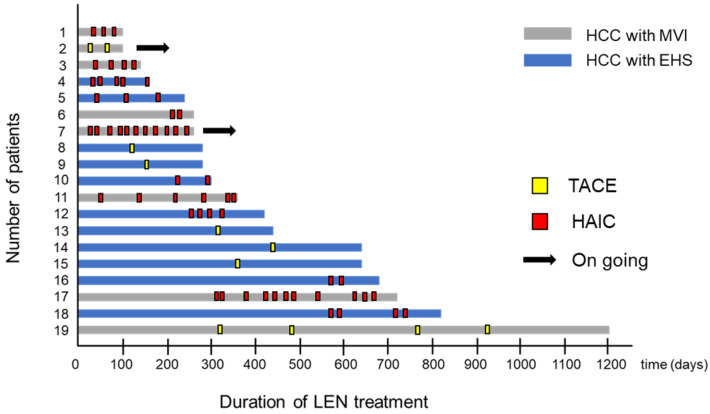
Swimmer plot analysis in the LEN combined with the TIT group. The point in the yellow square represents the added TACE. The point of the red square represents the added HAIC. The arrow indicates the continuation of treatment with LEN combined with TIT.

**Figure 4 ijms-24-13715-f004:**
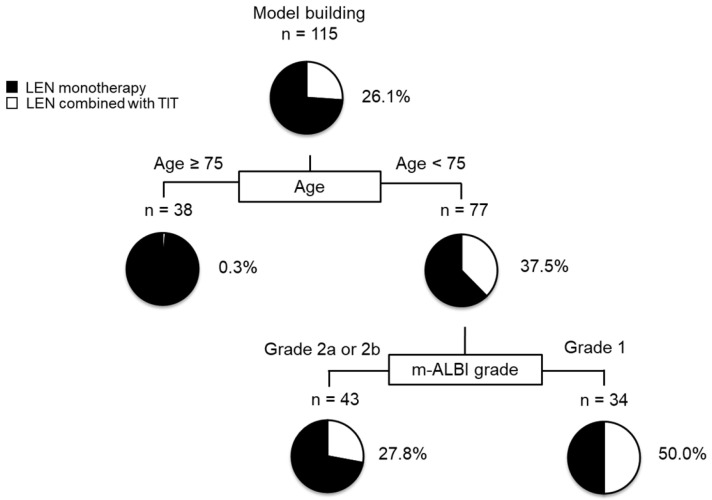
Profiles associated with the combination of LEN and TIT in patients with HCC treated with LEN. Decision-tree algorithm for LEN combined with TIT. The pie graphs indicate the percentage of LEN combined with TIT (white)/no-LEN combined with TIT (black) in each group. Profiles associated with the combination of LEN and TIT in patients with HCC treated with LEN. Decision-tree algorithm for LEN combined with TIT. The pie graphs indicate the percentage of LEN combined with TIT (white)/no-LEN combined with TIT (black) in each group.

**Table 1 ijms-24-13715-t001:** Patient characteristics.

Characteristic	All Patients	LEN Combined with TIT	LEN Monotherapy	*p*
*n*	115	30	85	
Age (years old)	71 (38–89)	63 (38–78)	73 (44–89)	<0.001
Sex (female/male)	23/92	5/25	18/67	0.590
PS (0/1/2)	105/8/1	28/2/0	77/7/1	0.475
Etiology (HBV/HCV/Others)	27/46/42	9/8/13	18/38/29	0.205
ALBI score (Median (range))	−2.50 (−3.34–−1.49)	−2.66 (−3.34–−1.91)	−2.47 (−3.33–−1.49)	0.001
m-ALBI grade (1/2a/2b)	47/34/34	18/6/6	29/28/28	0.047
Tumor size (mm)	40 (11–190)	40 (11–190)	40 (10–135)	0.396
Number tumors <5/≥5	31/84	8/22	23/62	0.966
Macrovascular invasion (No/Yes)	87/28	18/12	69/16	0.024
Extrahepatic spread (No/Yes)	27/88	10/20	17/68	0.138
Initial dose of LEN				0.454
12 mg	31	10	21	
8 mg	71	15	56	
8 mg weekend methods	3	2	1	
4 mg	8	2	6	
4 mg weekend methods	2	1	1	
Combination therapy				n.s.
TACE	12	12		
HAIC	18	18		
AFP (ng/mL)	92.7 (1.0–146,260)	78.9 (1.7–56,225)	133.0 (1.0–146,260)	0.605
Treatment line (1st/2nd/3rd)	77/27/11	21/6/3	56/21/8	0.869

Data are expressed as a median (range) or number. Abbreviations: LEN, lenvatinib; TIT; Transcatheter intra-arterial therapies; PS, Performance Status; HBV, hepatitis B virus; HCV, hepatitis C virus; ALBI score, albumin-bilirubin score; TACE, transcatheter arterial chemoembolization; HAIC, hepatic arterial infusion chemotherapy; AFP, α-fetoprotein.

**Table 2 ijms-24-13715-t002:** Patient characteristics after propensity score matching.

Characteristic	All Patients	LEN Combined with TIT	LEN Monotherapy	*p*
*n*	38	19	19	
Age (years old)	67 (44–79)	68 (44–78)	66 (44–79)	0.578
Sex (female/male)	8/30	4/15	4/15	n.s.
PS (0/1/2)	33/5/0	17/2/0	16/3/0	0.305
Etiology (HBV/HCV/Others)	12/11/15	5/5/9	7/6/6	0.597
ALBI score (Median (range))	−2.66 (−3.33–−1.91)	−2.69 (−3.06–−1.91)	−2.65 (−3.33–−2.10)	0.883
m-ALBI grade (1/2a/2b)	22/8/8	11/3/5	11/5/3	0.603
Tumor size (mm)	38 (11–131)	37 (11–131)	40 (11–92)	0.770
Number tumors <5/≥5	9/29	5/14	4/15	0.702
Macrovascular invasion (No/Yes)	25/13	11/8	14/5	0.303
Extrahepatic spread (No/Yes)	12/26	7/12	5/14	0.484
Initial dose of LEN				0.338
12 mg	12	5	7	
8 mg	23	11	12	
8 mg weekend methods	1	1	0	
4 mg	1	1	0	
4 mg weekend methods	1	1	0	
Combination therapy				n.s.
TACE	7	7		
HAIC	12	12		
AFP (ng/mL)	59.6 (1.6–56,225)	78.9 (1.7–56,225)	133.0 (1.0–146,260)	0.605
Treatment line (1st/2nd/3rd)	31/4/3	15/3/1	16/1/2	0.492

Data are expressed as a median (range) or number. Abbreviations: LEN, lenvatinib; TIT; Transcatheter intra-arterial therapies; PS, Performance Status; HBV, hepatitis B virus; HCV, hepatitis C virus; ALBI score, albumin-bilirubin score; TACE, transcatheter arterial chemoembolization; HAIC, hepatic arterial infusion chemotherapy; AFP, α-fetoprotein.

**Table 3 ijms-24-13715-t003:** Best response rate according to modified-RECIST after PSM.

Variables	LEN Combined with TIT (*n* = 19)	LEN Monotherapy (*n* = 19)	*p*
CR	1 (5.3%)	0 (0.0%)	
PR	8 (42.1%)	3 (15.7%)	
SD	9 (47.3%)	9 (47.4%)	
PD	1 (5.3%)	7 (36.9%)	
ORR	9 (47.3%)	3 (15.7%)	0.032
DCR	18 (94.7%)	12 (63.1%)	0.017

Abbreviations: RECIST, Response Evaluation Criteria in Solid Tumors; CR, complete response; PR, partial response; SD, stable disease; PD, progressive disease; ORR, objective response rate; DCR, disease control rate.

**Table 4 ijms-24-13715-t004:** Univariate and multivariate analyses of factors associated with OS after propensity score matching.

Variable	Univariate Analysis	Multivariate Analysis		
	*p*-Value	Odds Ratio	95% CI	*p*-Value
Age, <75 vs. ≥75	0.998			
Sex, female vs. male	0.226			
Etiology, (HBV vs. HCV vs. other)	0.245			
ALBI grade, 1 vs. 2	0.566			
Number tumors, <5 vs. ≥5	0.681			
Tumor size, <30 vs. ≥30	0.032	0.475	0.184–1.22	0.122
Macrovascular invasion, (No/Yes)	0.269			
Extrahepatic spread, (No/Yes)	0.495			
LEN combined with TIT, (+/−)	0.013	0.371	0.164–0.837	0.013
AFP, <200 vs. ≥200 ng/ml	0.001	0.367	0.161–0.832	0.012

Abbreviations: HBV, hepatitis B virus; HCV, hepatis C virus; ALBI grade, albumin-bilirubin grade; LEN, lenvatinib; TIT; Transcatheter intra-arterial therapies; AFP, α-fetoprotein.

## Data Availability

Data that support the findings of this study are available from the author, S.S. (Shigeo Shimose), on reasonable request.

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
