# Peer review of "Efficacy of Lenvatinib Combined with Transcatheter Intra-Arterial Therapies for Patients with Advanced-Stage of Hepatocellular Carcinoma: A Propensity Score Matching"

_ijms, 2023, doi:10.3390/ijms241813715_

Round 1
Reviewer 1 Report
In this study, Shigeo Shimose et. al. enrolled 115 patients with advanced-stage HCC and categorized them into the LEN combined with TIT group or the LEN monotherapy group. The authors evaluated the effect of lenvatinib (LEN) combined with transcatheter 22 intra-arterial therapy (TIT) for advanced-stage hepatocellular carcinoma (HCC) before and after propensity 23 score matching (PSM). The authors found that the OS in the LEN combined with TIT groups was significantly higher than that in the LEN monotherapy group and concluded that LEN combined with TIT may improve prognosis compared with LEN monotherapy in patients with advanced stage HCC.
Overall, the finding is interesting and has some novelty.
Comments:
Although the authors discussed the molecular mechanism underlying the treatment of LEN combined with TIT has a higher OS than LEN alone, if the authors can provide some experiment evidence, the results and conclusion will be more convincing.
Author Response
To REVIEWER 1
Thank you very much for your letter regarding our manuscript (ijms-2550283). We appreciate your comments, which have helped us to improve our manuscript. In line with your comments, please find below our point-by-point responses.
- Although the authors discussed the molecular mechanism underlying the treatment of LEN combined with TIT has a higher OS than LEN alone, if the authors can provide some experiment evidence, the results and conclusion will be more convincing.
Answer: We thank you for your valuable comment. It is well known that lnevtinib (LEN) improves drug delivery of anticancer drugs within the tumor through improving the intratumoral microenvironment of HCC tumors by normalizing blood vessels and reducing interstitial pressure [1, 2]. However, there are many reports regarding the induction of acquired resistance and various genetic changes in cancer cells in LEN alone treatment [3, 4]. we previously reported the acquired resistant mechanism of LEN due to the insulin-like growth factor-binding protein-1 (IGFBP-1) protein [5]. IGFBP1 is promoted by LEN-induced ischemia and it could induce neo-angiogenesis via VEGF-independent manner. In this way, acquired resistance can be induced by various reasons in LEN treatment. Therefore, transcatheter intra-arterial therapy (TIT) with improved drug delivery of anticancer drugs by LEN may directly and more effectively treat remnant LEN-resistant cancer cells.
However, transcatheter intra-arterial therapy (TIT) increases tumor hypoxia, thereby leading to the upregulation of hypoxia-inducible factor-1 (HIF-1). Elevated HIF-1 upregulates the expression of VEGF and PDGF, promoting tumor revascularization and progression [6, 7]. To solve these problems, re-administration of LEN after TIT prevents the recurrence through the control of the residual tumor and improvement of the hypoxic condition, resulting in the inhibition of releasing hypoxia-inducible cytokines [2]. Therefore, the synergistic effect of LEN and TIT may have improved OS compared to LEN alone.
As you indicated, we could not provide enough experiment evidence. Further study should focus on experimental evidence to build a more robust of the efficacy of TIT. These descriptions were added to the discussion and limitation sessions (Page: 9, Lines: 219-233; Page: 9, lines: 263). Once again, we deeply appreciate your comment, which greatly improved our manuscript.
Reference
- Carmeliet, P., and R. K. Jain. "Molecular Mechanisms and Clinical Applications of Angiogenesis." Nature 473, no. 7347 (2011): 298-307.
- Une, N., M. Takano-Kasuya, N. Kitamura, M. Ohta, T. Inose, C. Kato, R. Nishimura, H. Tada, S. Miyagi, T. Ishida, M. Unno, T. Kamei, and K. Gonda. "The Anti-Angiogenic Agent Lenvatinib Induces Tumor Vessel Normalization and Enhances Radiosensitivity in Hepatocellular Tumors." Med Oncol 38, no. 6 (2021): 60.
- Hou, W., B. Bridgeman, G. Malnassy, X. Ding, S. J. Cotler, A. Dhanarajan, and W. Qiu. "Integrin Subunit Beta 8 Contributes to Lenvatinib Resistance in Hcc." Hepatol Commun 6, no. 7 (2022): 1786-802.
- Zhou, H. M., J. G. Zhang, X. Zhang, and Q. Li. "Targeting Cancer Stem Cells for Reversing Therapy Resistance: Mechanism, Signaling, and Prospective Agents." Signal Transduct Target Ther 6, no. 1 (2021): 62.
- Suzuki, H., H. Iwamoto, T. Seki, T. Nakamura, A. Masuda, T. Sakaue, T. Tanaka, Y. Imamura, T. Niizeki, M. Nakano, S. Shimose, T. Shirono, Y. Noda, N. Kamachi, M. Sakai, K. Morita, M. Nakayama, T. Yoshizumi, R. Kuromatsu, H. Yano, Y. Cao, H. Koga, and T. Torimura. "Tumor-Derived Insulin-Like Growth Factor-Binding Protein-1 Contributes to Resistance of Hepatocellular Carcinoma to Tyrosine Kinase Inhibitors." Cancer Commun (Lond) 43, no. 4 (2023): 415-34.
- Sergio, A., C. Cristofori, R. Cardin, G. Pivetta, R. Ragazzi, A. Baldan, L. Girardi, U. Cillo, P. Burra, A. Giacomin, and F. Farinati. "Transcatheter Arterial Chemoembolization (Tace) in Hepatocellular Carcinoma (Hcc): The Role of Angiogenesis and Invasiveness." Am J Gastroenterol 103, no. 4 (2008): 914-21.
- Shim, J. H., J. W. Park, J. H. Kim, M. An, S. Y. Kong, B. H. Nam, J. I. Choi, H. B. Kim, W. J. Lee, and C. M. Kim. "Association between Increment of Serum Vegf Level and Prognosis after Transcatheter Arterial Chemoembolization in Hepatocellular Carcinoma Patients." Cancer Sci 99, no. 10 (2008): 2037-44.

Reviewer 2 Report
The authors tried to evaluate the effect of lenvatinib (LEN) combined with transcatheter intra-arterial therapy (TIT) for advanced-stage hepatocellular carcinoma (HCC) after propensity score matching (PSM). It is a retrospective study, unfortunately, but nevertheless, the study is highly relevant currently. They enrolled patients with advanced-stage HCC who received LEN treatment. They concluded that LEN combined with TIT may improve prognosis compared with LEN monotherapy in patients with advanced-stage HCC. I would suggest that the authors include new statements from the European Society of the Study of the Liver as well as similar statements from NASLD. The authors should expand the discussion by targeting the epi transcriptome. What is the role of LEN +TIT over LEN alone if we consider the cancer stem cells? Please answer properly and add a paragraph with regard to this topic.
Some sentences can be improved, but otherwise minor mistakes.
Author Response
To REVIEWER 2
Thank you very much for your letter regarding our manuscript (ijms-2550283). We appreciate your comments, which have helped us to improve our manuscript. In line with your comments, please find below our point-by-point responses.
- The authors should expand the discussion by targeting the epi transcriptome. What is the role of LEN +TIT over LEN alone if we consider the cancer stem cells? Please answer properly and add a paragraph with regard to this topic.
Answer: There are many reports regarding the induction of acquired resistance in lenvatinib (LEN) treatment. Anti-angiogenic drugs including LEN are generally effective for the proliferating cancer cells depending on the blood supply. However, cancer stem cells can survive an ischemic change due to anti-angiogenic drugs because they are independent of the blood supply [1]. Additionally, cancer stem cells are capable of self-renewal and multipotent differentiation. Recently, cancer stem cells have been shown to contribute to vasculogenic mimicry, which is a newly defined pattern of tumor blood supply[2]. Vasculogenic mimicry provides a special passage without endothelial cells and is conspicuously different from VEGF-inducing angiogenesis and vasculogenesis. Additionally, LEN can genetically induce various changes in cancer cells. Guo et al. demonstrated that YRDC, which was capable of binding transfer-RNA (tRNA), was related to acquired resistance to LEN by regulation of the protein translation of KRAS through the modification of tRNA[3]. Wei et al. also demonstrated that integrin subunit beta 8 (ITGB8) was a critical contributor to LEN-acquired drug resistance through the phosphorylation of HSP90[4]. Finally, we also reported the acquired resistant mechanism of LEN due to the insulin-like growth factor-binding protein-1 (IGFBP-1) protein[5]. IGFBP1 is promoted by LEN-induced ischemia and it could induce neo-angiogenesis via VEGF-independent manner. Taken together, acquired resistance can be induced by various reasons in LEN treatment. Therefore, the transcatheter intra-arterial therapy may directly and more effectively treat remnant LEN-resistant cancer cells. These descriptions were added to the revised manuscript (Page 8, Lines 207-212, Page 9, Line 213-233). Again, we appreciate your valuable comment, which has helped us to improve our manuscript.
Reference
- Zhou, H. M., J. G. Zhang, X. Zhang, and Q. Li. "Targeting Cancer Stem Cells for Reversing Therapy Resistance: Mechanism, Signaling, and Prospective Agents." Signal Transduct Target Ther 6, no. 1 (2021): 62.
- Yao, X. H., Y. F. Ping, and X. W. Bian. "Contribution of Cancer Stem Cells to Tumor Vasculogenic Mimicry." Protein Cell 2, no. 4 (2011): 266-72.
- Guo, J., P. Zhu, Z. Ye, M. Wang, H. Yang, S. Huang, Y. Shu, W. Zhang, H. Zhou, and Q. Li. "Yrdc Mediates the Resistance of Lenvatinib in Hepatocarcinoma Cells Via Modulating the Translation of Kras." Front Pharmacol 12 (2021): 744578.
- Hou, W., B. Bridgeman, G. Malnassy, X. Ding, S. J. Cotler, A. Dhanarajan, and W. Qiu. "Integrin Subunit Beta 8 Contributes to Lenvatinib Resistance in Hcc." Hepatol Commun 6, no. 7 (2022): 1786-802.
- Suzuki, H., H. Iwamoto, T. Seki, T. Nakamura, A. Masuda, T. Sakaue, T. Tanaka, Y. Imamura, T. Niizeki, M. Nakano, S. Shimose, T. Shirono, Y. Noda, N. Kamachi, M. Sakai, K. Morita, M. Nakayama, T. Yoshizumi, R. Kuromatsu, H. Yano, Y. Cao, H. Koga, and T. Torimura. "Tumor-Derived Insulin-Like Growth Factor-Binding Protein-1 Contributes to Resistance of Hepatocellular Carcinoma to Tyrosine Kinase Inhibitors." Cancer Commun (Lond) 43, no. 4 (2023): 415-34.
